# ArgBench: A Lean based Benchmark for Automated Theorem Provers on General-Purpose Reasoning Tasks

## Abstract

Recent advances in reinforcement learning–based large-model theorem provers have demonstrated remarkable progress in formal mathematical proof. However, their capabilities in broader formal reasoning tasks remain unclear. To address this gap, we introduce ArgBench, a benchmark dataset grounded in formal argumentation theory, designed to systematically evaluate large models on key abilities such as novel concept understanding and counterexample construction.

Our main contributions are as follows. First of all, we select formal argumentation theory—a relatively underexplored domain in logic with many open problems—which substantially reduces the risk of pretraining data leakage or contamination and enables a more faithful assessment of models' capacity to adapt to new definitions and rules. Secondly, we propose a type-theoretic automatic generation method that constructs large-scale datasets at minimal human cost. Thirdly, the generation algorithm is decoupled from any specific domain, allowing straightforward transfer to other formal reasoning settings.

Evaluation on ArgBench reveals that mainstream large-model provers perform poorly overall, with Goedel Prover achieving only a 5.7% success rate. Further analysis highlights a particular weakness in counterexample construction. Based on these findings, we suggest a promising direction: using ArgBench as a training environment to enhance counterexample construction through reinforcement learning, thereby advancing toward more general-purpose formal reasoning.

## 1 Introduction

LLMs combined with proof assistants (e.g., Lean/Isabelle) have rapidly advanced machine-checkable reasoning, aided by autoformalization, reinforcement learning (RL), and agentic orchestration (Polu & Sutskever, 2020; Zheng et al., 2021; Lin et al., 2025; Wang et al., 2025; Ren et al., 2025; Zhou et al., 2025; Chen et al., 2025). State-of-the-art systems attain strong scaling on miniF2F and related suites, either by step-level interaction with search or by whole-proof generation augmented with long reasoning traces and verifier feedback. While this progress is notable, most evaluations remain *in-distribution*: models operate within well-known libraries and fixed logical infrastructure, raising the question of how well they *generalize to new formal rules and definitions* introduced only at test time.

We address this gap with **ArgBench**, which frames evaluation in the niche but expressive setting of *formal argumentation*. In Dung-style argumentation frameworks (AFs), nodes are arguments and directed edges encode attacks; semantics (e.g., grounded, preferred, stable) specify which sets of arguments are acceptable under conflict and defense (Dung, 1995). This domain is appealing for three reasons: (i) it is comparatively niche, reducing pretraining contamination; (ii) solving tasks requires *semantic uptake*—the model must internalize freshly provided definitions (attack, conflict-freeness, defense) and apply a specified semantics; and (iii) the reasoning patterns abstract to transferable procedures (e.g., fixpoint computations on graphs) that recur across formal systems (verification, modal/temporal logics). In spirit, ArgBench complements recent FTP pipelines that scale via verifiable signals, curriculum/subgoal decomposition, and domain-aware orchestration (Wang et al., 2025; Ren et al., 2025; Lin et al., 2025; Zhou et al., 2025; Chen et al., 2025).

**Contributions.** This work introduces **ArgBench**, a benchmark for automated theorem proving (ATP) in abstract argumentation, a specific logic for AI, and advances the field along three axes:

**Type-theoretic generation algorithm** We cast instance synthesis and verification in a lightweight, curried type-theoretic framework (Pierce, 2002; Harper, 2016; Church, 1940; Curry & Feys, 1958). This design yields seed-data independence (instances arise whenever some object has tail type Prop) (Coquand & Huet, 1988; Luo, 1990; Howard, 1980; Girard et al., 1989; Martin-Löf, 1984). , a purely algorithmic and scalable pipeline with difficulty tunable via $C_n \times D_m$, substantial instance diversity despite simple construction rules, and calibrated correctness through an LEM pairing that fixes the valid/invalid balance at $50\%$.

**Abstract argumentation as the seed domain.** We deliberately adopt abstract argumentation to mitigate data contamination—most generated queries are open-form without canonical published solutions likely memorized during pretraining—and to catalyze progress in an underexplored logical area, where large-scale AI triage can surface difficult subcases and focus expert attention.

**Empirical evaluation and diagnostic.** We evaluate three state-of-the-art provers on ArgBench and identify a central failure mode: insufficient robustness in *counterexample construction*; this suggests incorporating targeted counterexample signals into future reinforcement learning or fine-tuning pipelines to strengthen semantics-conditioned generalization.

## 2 RELATED WORK

**LLMs for Formal Theorem Proving (FTP).** Early neural provers demonstrated that large LMs can emit tactic sequences or whole Lean scripts but struggled on Olympiad-level math (Polu & Sutskever, 2020; Zheng et al., 2021). Rapid progress in 2024–2025 came from two directions: (i) *whole-proof* generation with RL or curriculum, and (ii) *agentic* orchestration with verifier-in-the-loop. On the RL side, Wang et al. (2025) train a 72B "formal reasoning pattern" achieving strong scaling and state-of-the-art miniF2F scores; Ren et al. (2025) unify informal decomposition with subgoal curriculum, reporting top results on miniF2F and a new ProverBench. Data-centric scaling also matters: Lin et al. (2025) autoformalize and bootstrap massive Lean corpora to train Goedel-Prover, substantially improving pass rates on miniF2F and PutnamBench (Tsoukalas et al., 2024). On the agentic side, Zhou et al. (2025) show that a general-purpose LLM inside a reflective, decomposition-and-repair loop can rival bespoke provers; Chen et al. (2025) adopt lemma-style whole-proof reasoning and a specialized geometry backend to nearly saturate miniF2F while tackling IMO-grade problems. Tooling such as LeanDojo streamlines retrieval-augmented proving and programmatic access to Lean kernels (Yang et al., 2023). These advances echo broader trends in scaling and representation learning (Bengio & LeCun, 2007; Hinton et al., 2006; Goodfellow et al., 2016).

**Benchmarks for Formal Reasoning and Generalization.** MiniF2F (Zheng et al., 2021) remains a central Lean benchmark; ProofNet (Azerbayev et al., 2023) couples informal/formal pairs to evaluate autoformalization; PutnamBench (Tsoukalas et al., 2024) expands contest coverage. General reasoning suites (e.g., BIG-bench and GPQA) emphasize difficult, contamination-resistant questions (Srivastava et al., 2022; Rein et al., 2023), and dynamic testbeds like LiveBench refresh tasks to mitigate leakage (White et al., 2025). Still, as provers approach saturation on familiar distributions, evaluating *generalization to new definitions and rules* becomes crucial—precisely the gap our argumentation-based benchmark targets.

**Autoformalization and Informal–Formal Bridging.** LM-powered autoformalization has improved steadily, from code-model pipelines to retrieval- and checker-in-the-loop refinement (Wu et al., 2022; Azerbayev et al., 2023). Natural-language proof datasets and reasoners (e.g., Natural-Proofs, ProofWriter) probe the capacity to structure arguments without a proof assistant (Welleck et al., 2021; Tafjord et al., 2021). Recent FTP systems increasingly incorporate such bridges (e.g., informal decomposition guiding formal subgoals (Ren et al., 2025)), suggesting a convergent path where informal reasoning scaffolds formal synthesis.

**Agentic Orchestration, Tools, and ATP Hybrids.** From early RL-in-the-assistant environments (HOList/DeepHOL, CoqGym) (Bansal et al., 2019; Yang & Deng, 2019) to modern SWE-style

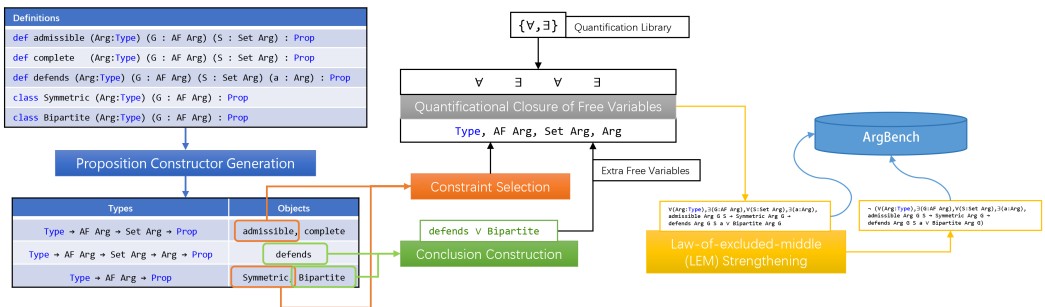

Figure 1: An overview of our framework

agent frameworks, test-time tool use improves sample efficiency and robustness. Verifier feedback, retrieval, and domain backends (e.g., geometry engines) are increasingly standard in frontier provers (Chen et al., 2025). More generally, scaling laws for reasoning benefit from structure-inducing supervision (e.g., chain-of-thought prompting (Wei et al., 2022)) and verifiable signals, while strong base models (e.g., GPT-4) exhibit "sparks" of advanced reasoning yet still rely on orchestration for reliability (Bubeck et al., 2023; OpenAI, 2023). Foundational infrastructures (Lean 4; Isabelle/HOL) anchor this progress with fast kernels and mature libraries (de Moura & Ullrich, 2021; Nipkow et al., 2002).

## 3 METHODOLOGY

**Principle-based methodology.** Principle-based methodologyYu et al. (2021) is a prominent line of inquiry in argumentation theoryAlfano et al. (2024); Amgoud & Vesic (2011). It examines the satisfiability of different principles across classes of argumentation frameworksFazzinga et al. (2022); Bonzon et al. (2016). More generally, given $n$ classes of frameworks and $m$ target conclusions, one considers an $n \times m$ table that records, for each pair, whether the conclusion is satisfiable within that class. The entry in row $i$, column $j$ indicates the satisfiability of the $j$-th conclusion for the $i$-th framework class. Each cell takes one of two values: **fully satisfiable** or **not fully satisfiable**. The former means the conclusion holds uniformly across all frameworks in that class (i.e., for every instance of class $i$, the $j$-th conclusion is true); the latter means there exists at least one framework in class $i$ that violates the $j$-th conclusion.

**Our approach.** Our dataset-generation procedure is inspired by, and extends, this principle-based perspective. At core, an argumentation framework is a specialized graph endowed with framework-level properties; distinctions between framework classes reduce to differences in data structure and in these properties. Because data and constraints are coupled, we cannot, as in propositional logic, freely permute premises to synthesize new instances. To enable flexible instance construction, we first decouple *logic* from *data*. Concretely, we *generate the logic first* and *instantiate the data second*: we (i) construct, without restrictions, proposition-forming constructors whose *tail type* is Prop as candidate constraints, (ii) supply the free variables required by these constructors, and (iii) convert those free variables into constrained (bound) variables, thereby closing the instance.

### 3.1 PRELIMINARIES

#### 3.1.1 TAIL TYPE

We work in the simply typed $\lambda$-calculus (STLC) with curried arrow types, where arrows associate to the right.

**Type layer (STLC).** Types are generated by the grammar

$$\tau ::= \alpha \mid \tau \to \tau,$$

where $\alpha$ ranges over base types (e.g., Bool, Nat, or a structured type AF). By convention, $A \to B \to C \equiv A \to (B \to C)$.

**Definition 1 (Tail type)** *For any type $\tau$, define its* tail type $\mathrm{tail}(\tau)$ *inductively by*

$$\mathrm{tail}(\alpha) \triangleq \alpha, \qquad \mathrm{tail}(\sigma \to \rho) \triangleq \mathrm{tail}(\rho).$$

*Equivalently, view $\tau$ as a binary tree whose internal nodes are "$\to$" and whose leaves are (base or non-arrow) types; then $\mathrm{tail}(\tau)$ is the rightmost leaf of this tree. In particular, $\mathrm{tail}(A_1 \to \cdots \to A_n \to R) = R$.*

**Term layer.**   Suppose $\Gamma \vdash t : \tau$ with $\tau \equiv A_1 \to \cdots \to A_n \to R$. For any arguments $\Gamma \vdash a_i : A_i$, consider the full application

$$t \, a_1 \cdots a_n \ \xrightarrow{\beta^*} \ v,$$

where $v$ is in $\beta$-normal form. By subject reduction and progress,

$$\Gamma \vdash t \, a_1 \cdots a_n : R \quad \text{and} \quad \Gamma \vdash v : R,$$

hence $R = \mathrm{tail}(\tau)$. In words: *the result type of a fully applied term, after $\beta$-reduction to normal form, coincides with the tail type of its original type.*

**Example 1** *If $\tau = A \to (B \to C)$, then $\mathrm{tail}(\tau) = C$. If $\Gamma \vdash t : \tau$, $\Gamma \vdash a : A$, and $\Gamma \vdash b : B$, then $t \, a \, b \xrightarrow{\beta^*} v$ with $\Gamma \vdash v : C$.*

Intuitively, the *tail type* is the residual codomain after stripping all left-hand arrow binders (formal parameters) from a curried type; equivalently, it is the result type obtained by fully applying any term of that type and reducing to $\beta$-normal form.

**A Remark.** We *define* the tail type $\mathrm{tail}(\tau)$ **at the type level** as the rightmost codomain of a (right-associated) curried arrow type; **at the term level** we prove that the $\beta$-normal result of a fully applied term of type $\tau$ has type $\mathrm{tail}(\tau)$; for dependent products we write $\mathrm{tail}^\Pi$ (with explicit argument indices) to denote the argument-dependent rightmost codomain—this is a notational refinement of the same type-level notion, *not* a separate concept.

Our dataset-generation pipeline consists of five sequential stages: *proposition constructor generation*, *constraint selection*, *conclusion construction*, *quantificational closure of free variables*, and *law-of-excluded-middle (LEM) strengthening*. We'll go over each stage in detail below.

## 3.2   Proposition Constructor Generation

The aim of this stage is to identify all *candidate proposition constructors*, namely terms whose tail type is Prop. A *proposition constructor* is any term $c : \tau$ with $\mathrm{tail}(\tau) = \mathsf{Prop}$. For example,

$$\lambda(x : \mathsf{N}). \, x = 0 \ : \ \mathsf{N} \to \mathsf{Prop}$$

is a proposition constructor: given a natural number, it yields the proposition asserting that the input equals $0$.

Accordingly, we scan the available primitive objects (constants and closed terms) and perform a tail-type check. Via a simple structural recursion on types, we collect into a set $S$ all objects whose tail type is Prop; these constitute the pool of candidate proposition constructors.

## 3.3   Constraint Selection

In this step, we choose a subset $C \subseteq S$ of the candidate proposition constructors to serve as assumptions (constraints). By logical monotonicity, for any $C_1, C_2 \subseteq S$, if $C_1 \subseteq C_2$ and a conclusion is derivable from $C_1$, then it is also derivable from $C_2$. Hence, using more assumptions weakly simplifies derivability.

Assume $S$ contains $n$ proposition constructors. Let

$$\boldsymbol{C}_n \coloneqq \{\, C \subseteq S \mid |C| = n \,\}$$

denote the family of size-$n$ subsets of $S$. In the experiments reported in this paper, we set $n = 2$.

## 3.4 CONCLUSION CONSTRUCTION

In general, we regard a conclusion as a compound proposition built from atomic propositions using logical connectives. Any compound proposition can be put into conjunctive normal form (CNF), i.e., as a conjunction of disjunctive clauses. Moreover, implication distributes over conjunction on the right,

$$A \to (B \wedge C) \quad \Longleftrightarrow \quad (A \to B) \wedge (A \to C),$$

so proving a disjunction *as a whole* under the same antecedent is equivalent to proving each conjunct separately. Consequently, it suffices to consider conclusions that are (conjunctive) clauses.

Conceptually, premises and conclusions are both just propositions; they differ only in their roles within an instance. We therefore construct conclusions from the same candidate pool used for constraints. Let $\boldsymbol{C}_n$ be the family of size-$n$ subsets of $S$. We define the candidate conclusion set

$$\boldsymbol{B}_m \ := \ \left\{ \bigvee_{a_i \in B} \overline{a_i} \ \middle| \ B \in \boldsymbol{C}_n \right\}, \quad \text{where} \quad \overline{a_i} \in \{a_i, \neg a_i\}.$$

In the experiments reported in this paper, to control the total number of conclusions, we fix $\overline{a_i} = a_i$ (i.e., we do not introduce negations).

## 3.5 QUANTIFICATIONAL CLOSURE OF FREE VARIABLES

In the final step, we select an element $(a, b) \in \boldsymbol{A}_n \times \boldsymbol{B}_m$ to form a (premise, conclusion) constructor pair $(C, D)$, and we discard *trivial* cases in which the conclusion syntactically contains one of the premises. At this stage, both the conclusion and the premises are still constructor states targeting Prop; therefore, we need to supply constrained variables and obtain propositions via $\beta$-reduction. To this end, we first extract the free variables (i.e., curried parameter types) of a constructor:

$$\mathsf{Free}(S) = \begin{cases} \varnothing & \text{if } S : \mathsf{Prop}, \\ \{A\} \cup \mathsf{Free}(B) & \text{if } S : A \to B. \end{cases}$$

Given the set of premises $C$ and the conclusion $D$, define

$$F \ := \ \left( \bigcup_{c \in C} \mathsf{Free}(c) \right) \cup \mathsf{Free}(D).$$

For each $A \in F$, introduce a fresh constrained variable $x_A{:}A$ and write $f(A) = x_A$ for the environment. We instantiate constructors by feeding these variables and performing $\beta$-reduction via the following recursive replacement:

$$R_f(S) = \begin{cases} S & \text{if } S : \mathsf{Prop}, \\ R_f(S \, x_A) & \text{if } S : A \to B \text{ and } A \in F. \end{cases}$$

This yields instantiated premises and conclusion

$$C^* := \{ R_f(c) \mid c \in C \}, \qquad D^* := R_f(D),$$

and the unclosed implication

$$\Phi \ := \ \left( \bigwedge_{\psi \in C^*} \psi \right) \ \to \ D^*.$$

Next, we close all constrained variables with quantifiers. Let $\Sigma := \{\sigma : F \to \{\forall, \exists\}\}$ and $\Pi := S_{|F|}$ be the full permutation group. Since the presence of existential quantifiers breaks commutativity of quantifier prefixes, we enumerate permutations only when necessary:

$$Q(F) \ := \ \left\{ \mathsf{Quant}(\pi, \sigma) \ \middle| \ \sigma \in \Sigma, \ \pi \in \begin{cases} \{\mathrm{id}\} & \text{if } \forall A \in F, \ \sigma(A) = \forall, \\ \Pi & \text{otherwise} \end{cases} \right\},$$

where the action of a quantifier prefix on a formula $\varphi$ is (for a fixed ordering $F = \{A_1, \ldots, A_n\}$)

$$\mathsf{Quant}(\pi, \sigma)[\varphi] := \big(\sigma(A_{\pi(1)})\, x_{A_{\pi(1)}}{:}A_{\pi(1)}\big) \cdots \big(\sigma(A_{\pi(n)})\, x_{A_{\pi(n)}}{:}A_{\pi(n)}\big)\, \varphi.$$

Therefore, the candidate set of closed formulas associated with $(C, D)$ is

$$\mathcal{P}(C, D) := \big\{\, \mathsf{Quant}(\pi, \sigma)[\Phi] \;\big|\; \mathsf{Quant}(\pi, \sigma) \in Q(F) \,\big\}.$$

To remove trivialities, define a decision function $\mathrm{Trivial}(\cdot)$: if there exists $\psi \in C^*$ such that, after allowing $\alpha$-renaming and restricted $\beta\eta$-equivalences, $D^*$ contains $\psi$ as a syntactic subformula, then the formula is labeled trivial. The final output is

$$\mathcal{P}_{\mathrm{nontriv}}(C, D) := \big\{\, \varphi \in \mathcal{P}(C, D) \;\big|\; \neg\, \mathrm{Trivial}(\varphi) \,\big\}.$$

*Remark.* In a dependent type setting, the relative order of quantifiers and parameters must respect dependency; the above enumeration of $Q(F)$ implicitly assumes a simple (non-dependent) type discipline and should be adapted to dependency-aware permutations when needed. In our experiments, we adopt a simple setting that only considers universal quantifiers. Under this assumption, all quantifiers commute, so we keep a single canonical order (along the fixed ordering of $F$) and write

$$Q(F) = \forall x_{A_1}{:}A_1 \;\; \forall x_{A_2}{:}A_2 \;\; \cdots \;\; \forall x_{A_{|F|}}{:}A_{|F|},$$

each $x_{A_i}$ is the corresponding constrained variable.

We now encode the construction of a closed problem formula from a (conditions, conclusion) pair as a function $R$. Let

$$\Phi := \Big(\bigwedge_{c \in C} R_f(c)\Big) \rightarrow R_f(D),$$

then, in the universal-quantifier setting,

$$R(C, D) := Q(F)[\Phi] = \Big(\forall x_{A_1}{:}A_1 \;\cdots\; \forall x_{A_{|F|}}{:}A_{|F|}\Big)\Big(\bigwedge_{c \in C} R_f(c) \rightarrow R_f(D)\Big),$$

.

Finally, we apply $R$ to all pairs in $\boldsymbol{A}_n \times \boldsymbol{B}_m$ and remove the *trivial* cases in which the conclusion syntactically contains a condition, obtaining the set of formulas

$$P := \big\{\, R(C, D) \;\big|\; (C, D) \in \boldsymbol{A}_n \times \boldsymbol{B}_m,\; \neg\, \mathrm{Trivial}\big(R(C, D)\big) \,\big\}.$$

If existential quantifiers are introduced in future settings, one must restore the enumeration over quantifier assignments and (when necessary) over permutations of the quantifier prefix. In the present universal-only setting, this enumeration is unnecessary.

### 3.6 LAW-OF-EXCLUDED-MIDDLE (LEM) STRENGTHENING

However, formulas constructed in this way may be false or even unprovable. In argumentative settings, prior work suggests that known equivalences are a vanishingly small fraction of the enumerated search space; consequently, most of the formulas we generate are open, and we cannot tell whether a failure to prove a formula is due to its falsity or to the prover's limitations. To address this, we adopt a simple augmentation based on the Law of Excluded Middle (LEM). For every problem $p \in P$, we also include its negation, yielding the augmented set

$$\bar{P} = P \cup \{\, \neg p \mid p \in P \,\}.$$

Under classical, sound, and complete reasoning, exactly one of $p$ or $\neg p$ is valid (and thus provable), ensuring that precisely half of the instances in $\bar{P}$ are correct/provable. This provides a reliable denominator when computing proof success rates.

Figure 2: The Structure of the Counter Example

## 4 EXPERIMENTS

We take *structured abstract argumentation* as the base domain and extract $|S| = 17$ formula constructors from our formal code base. Using the generation setting $C_2 \times D_1$, we produce 858 condition–conclusion pairs, i.e., 1716 formulas in total (each pair yields a positive instance and its negation via the LEM augmentation). Aggregating these 858 pairs gives the benchmark we refer to as **ArgBench**.

We evaluate four state-of-the-art open-source automated theorem provers. The results are summarized in Table 1. "PassNum" counts the number of pairs for which a (positive or negative) proof was found, and "PassRate" is computed against the 858 pairs. "Positive Proof" and "Negative Proof" report, respectively, how many positive instances and negated instances were proved.

| Prover | PassNum | PassRate | Params | Samples | Positive | Negative |
|---|---|---|---|---|---|---|
| Goedel-Prover-V2 | 49 | 5.71% | 8B | 1 | 49 | 0 |
| Kimina-Prover-RL | 0 | 0.00% | 1.7B | 1 | 0 | 0 |
| Kimina-Prover-Distill | 0 | 0.00% | 7B | 1 | 0 | 0 |
| DeepSeek-Prover-V2 | 70 | 8.16% | 7B | 1 | 70 | 0 |

Table 1: Results on ArgBench (858 pairs / 1716 formulas). "PassRate" is PassNum divided by 858.

We observe that two provers exhibit nontrivial transfer: although designed primarily for mathematical problem solving, they can still prove a subset of our logical instances. However, none of the systems successfully prove *negative* instances. In principle, refuting a false statement can be straightforward—often a single counterexample suffices. Below we present a human-written refutation based on a concrete counterexample to illustrate the simplicity of such proofs for many negated instances.

```
1  def SimpleAF2:AF (Fin 2):={att:=fun x y =>(x=0∧y=1)∨(x=1 ∧y=0)}
2  def S02 : Set (Fin 2) := {x | x = 0}
3  def S0 : Set (Fin 2) := ∅
4  theorem test:¬(∀(Arg:Type),∀ (S:Set Arg),∀ (G:AF Arg),
5  AF.stable G S → AF.complete G S → AF.grounded G S ) := by
6  push_neg;use (Fin 2),S02,SimpleAF2;repeat constructor;
7  repeat simp[S02,SimpleAF2];
8  intros;omega;constructor;simp[S02,SimpleAF2];intro a b;by_contra ha
9  have ha:a=1:=by omega
10 rw [ha] at b;simp at b;by_contra h;have h1:= h.1
11 have h3: SimpleAF2.complete S0:= by
12   constructor;simp[SimpleAF2,S0];intro a h1;simp[S0]
13   by_cases h:a=0
14   rw [h] at h1;simp[SimpleAF2,S0] at h1;
15   have ha:a=1:= by omega
16   rw [ha] at h1;simp[SimpleAF2,S0] at h1
17 have h4:=(h.2 S0) h3;simp [S02,S0] at h4
```

Listing 1: Counterexample to Stable + Complete ⇒ Grounded

In this proof, we construct an exceptionally simple counterexample. We define the set $S_2 = \{0\}$ and the set $S_0 = \emptyset$. In the argumentation framework shown in Figure 2, $S_2$ is a **stable** and **complete** extension, but it is not a **grounded** extension, as the grounded extension is $S_0$. While classical automated theorem provers (ATPs), such as Nitpick, can find such counterexamples, current large language models do not yet possess this constructive capability.

An alternative route is to leverage Isabelle/HOL's NITPICK tool to rapidly search for counterexamples and thereby validate refutations. One can use NITPICK to auto-generate (counter)model witnesses for candidate formulas, then convert these witnesses into labeled instances to expand the dataset and train proof models with richer negative supervision.

## 5 CONCLUSION

Grounded in the formal problem of abstract argumentation, we introduce **ArgBench**, a benchmark for automated theorem proving, and evaluate it on three state-of-the-art (SOTA) models. The results indicate that, despite its seemingly simple construction, the dataset remains challenging for current systems.

This paper makes three primary contributions as follows:

### 5.1 A TYPE-THEORETIC METHOD FOR DATASET CONSTRUCTION

We propose a novel, type-theoretic approach to dataset construction and augmentation, with the following advantages:

1. **Seed-data independence.** The procedure imposes no requirements on seed data: as long as there exists an object whose tail type is Prop, instances can be synthesized. This confers strong generality, especially for new, open domains, enabling AI to rapidly explore unfamiliar territories and derive basic conclusions, thereby advancing AI4Science.

2. **Purely algorithmic pipeline.** No learned model is used for augmentation; the process is driven entirely by enumeration. Consequently, we can synthesize arbitrarily large datasets at very low cost and control difficulty by tuning the parameter $C_n \times D_m$. As model capabilities improve, the same knobs can be adjusted accordingly.

3. **Diversity of instances.** Although the generation procedure may appear to yield limited variety, experiments show that most synthesized conjectures are in fact unprovable. This reflects the inherently "chaotic" nature of mathematics: small perturbations to the premises can invalidate prior proofs and necessitate new ones.

4. **Correctness guarantees.** Our logical design yields reliable validity estimates. For instance, early datasets such as MiniF2F include misformalized items that are, in principle, unprovable; auto-formalized corpora suffer similar issues. By employing a law-of-excluded-middle (LEM) pairing technique, we guarantee that exactly $50\%$ of our instances are valid, enabling precise measurement of model ability.

### 5.2 ABSTRACT ARGUMENTATION AS THE SEED DOMAIN

As the seed problem, we deliberately adopt abstract argumentation, a comparatively niche area of logic, for two reasons:

1. **Mitigating data contamination.** Most generated questions are open problems without published solutions. In contrast to many standard mathematics problems with fixed answers that may have been seen during pretraining, our instances more cleanly probe model capability.

2. **Stimulating underexplored fields.** As automated proving improves, datasets constructed via our method can help catalyze progress in such domains. Large-scale AI triage can surface especially difficult subcases and efficiently direct expert attention.

### 5.3 EMPIRICAL ANALYSIS ON SOTA MODELS

Our experiments diagnose a central failure mode: poor performance largely stems from the lack of robust counterexample construction. We recommend incorporating targeted training signals for counterexample generation in future reinforcement learning or fine-tuning pipelines, so that models can adapt to a broader range of situations.

## 6  FUTURE WORK

There are several directions for strengthening our framework:

1. **Supporting multiple objects of the same type.** At present, variables of the same type collapse to a single representative. In principle, one can partition objects of a given type into equivalence classes; our current setting corresponds to the special case with a single class. Each class could then be constrained by distinct quantifiers.

2. **Supporting nested constructor chaining.** We currently instantiate only direct propositional constructors. More generally, heterogeneous objects can be composed via $\beta$-reduction to yield new propositional constructors.

3. **Supporting difficulty decomposition.** Our theory suggests that enlarging the premise set typically simplifies proofs. We can introduce a mechanism that, upon a proof failure, automatically strengthens the premises to reduce difficulty. If adding a pair of mutually exclusive conditions renders both branches provable, one effectively obtains a proof by case analysis of the original statement. This differs from commonly used "subgoal guessing" heuristics.

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
