# OpenReview forum: "ArgBench: A Lean based Benchmark for Automated Theorem Provers on General-Purpose Reasoning Tasks"
_ICLR.cc/2026/Conference — Submitted to ICLR 2026_

### Official Review · Reviewer_4eDL · 2025-10-25

**Soundness:** 3
**Presentation:** 2
**Contribution:** 2
**Rating:** 4
**Confidence:** 4

**Summary:**

This paper introduces ArgBench, a Lean-based benchmark intended to test generalization of ATPs beyond familiar math libraries. The core technical piece is a type-theoretic generation pipeline that identifies proposition constructors whose tail type is Prop, picks small subsets as premises, builds conclusions from the same pool, closes free variables with quantifiers, then adds LEM-paired negation so that exactly half of the instances are valid/provable.

The authors produce 858 condition–conclusion pairs (1716 formulas with LEM), and evaluate open-source provers (Goedel-Prover-V2 8B, DeepSeek-Prover-V2 7B, and two Kimina variants) using a single sample per problem. Reported pass rates are low (5.71% and 8.16% for Goedel-Prover-V2 and DeepSeek-Prover-V2), with 0 refutation proofs. The paper argues ArgBench could serve as a training environment to improve counterexample construction.

**Strengths:**

1. This paper introduces a clear formal dataset construction method in Lean with a principled notion of proposition constructors (tail-type to Prop), and an explicit five-stage pipeline, which is a promising for large-scale data generation.

2. The focused domain in this paper reduces data contamination and forces uptake of new definitions provided at test time. This targets a real gap in current ATP evaluation.

**Weaknesses:**

1. The key failure mode is refutation/counterexample construction, yet the current generator omits existentials and dependency-aware quantifiers that naturally produce witness-oriented tasks. LEM pairing guarantees a 50/50 label balance but does not by itself create refutable formulas that are constructively easy in Lean. A principled design for negative instances (e.g., by integrating Nitpick-derived finite AF witnesses into the instance space) is only discussed as future work, not executed. Since the paper only includes ~8 pages of contents out of limit of 9 pages and the appendix is not included, I encourage authors to extend the work by adding more explanations.

2. The source code / dataset files are not released in supplementary material and anonymous repo, making it difficult to reproduce the experiment, and making results of the paper unconvincing.

3. The title and claims contained overstated scope by emphasizing general-purpose reasoning, but all experiments are limited to one formalism of abstract argumentation and a narrow constructor set. The connection to general reasoning beyond graph-based fixpoint intuitions is not demonstrated.

**Questions:**

1. Regarding the sampling budges, have the authors tried higher sample numbers (i.e. higher n in Pass@n) to maximize model capability? Can you report results with prover setups (e.g., Pass@32) that aligns with most works on AI4Math like [1]? This would make the SOTA comparison commensurate.

2. Existentials and witnesses: Can you extend the generator to allow existentially quantified variables and dependency-aware permutations and show whether this alleviates the "no negative proofs" failure mode?

3. Could you please include difficulty curves as you vary $n,m$, and a breakdown by semantics (grounded/preferred/stable)?

[1] Ren, Z. Z., et al. "Deepseek-prover-v2: Advancing formal mathematical reasoning via reinforcement learning for subgoal decomposition." arXiv preprint arXiv:2504.21801 (2025).

---

### Official Review · Reviewer_ecr5 · 2025-10-31

**Soundness:** 2
**Presentation:** 3
**Contribution:** 2
**Rating:** 4
**Confidence:** 4

**Summary:**

This paper introduces ArgBench, a benchmark for evaluating automated theorem provers on formal argumentation tasks. The authors propose a type-theoretic generation method to automatically create theorem-proving instances from abstract argumentation frameworks (based on Dung's semantics: grounded, preferred, stable extensions). The generation pipeline has five stages: (1) identifying proposition constructors with tail type Prop, (2) selecting constraint subsets, (3) constructing conclusions, (4) closing free variables with quantifiers, and (5) using Law of Excluded Middle (LEM) to ensure 50% of instances are provable.

Using setting $C_2 \times D_1$​, they generate 858 premise-conclusion pairs (1,716 formulas total). Evaluation of four SOTA provers shows very poor performance: Goedel-Prover achieves only 5.7% success rate, DeepSeek-Prover-V2 reaches 8.16%, and notably,
no prover successfully proves any negative instances. The authors identify counterexample construction as a critical weakness and suggest using ArgBench as a training environment to improve this capability through reinforcement learning.

**Strengths:**

1. Originality: Applying LLM theorem provers to abstract argumentation is novel. Type-theoretic approach is more systematic than ad-hoc construction. Clever use of classical logic to ensure balanced dataset.

2. Quality: Mathematical framework is sound and well-defined. Listing 1 provides a clear illustration of the failure mode. Generation algorithm is described in sufficient detail.

3. Clarity: Clear delineation of the 5-stage pipeline. Formal definitions (tail type, etc.) are rigorous. Authors clearly state low performance results.

4. Significance: Counterexample construction is an underexplored weakness in current provers. Data could be used for targeted RL training on counterexample generation. Niche domain likely reduces memorization issues.

**Weaknesses:**

Major Issues

1. Scale limitations:

1.1 858 pairs is small for a benchmark claiming "large-scale" or "systematic evaluation".

1.2 Compare to: miniF2F (488 problems), ProofNet (371 pairs), PutnamBench (640 problems), even these are considered small.

1.3 No justification for why $C_2 \times D_1$​​ was chosen or why scaling was not explored.

1.4 With 17 constructors, $C_2$ yields only $\binom{17}{2} = 136$ constraint pairs, this limits diversity.

2. "General-Purpose Reasoning" claims:

2.1 Abstract argumentation is an niche subdomain of logic.

2.2 No evidence that performance here correlates with general reasoning ability.

2.3 The paper conflates "novel domain" with "general-purpose".

2.4 Title and abstract promise general reasoning evaluation but deliver narrow domain testing.

3. Counterexample analysis:

3.1 Core finding (0% negative proofs) is striking but underanalyzed.

3.2 Are all negative instances equally hard? What makes them difficult?

3.3 No error analysis or failure mode categorization.

3.4 No comparison to classical counterexample finders (Nitpick is mentioned but not compared).

3.5 No attempt to generate counterexamples separately and test if models can verify them.

4. Experimental Evaluation:

4.1 Single sample per model (n=1) with no statistical analysis.

4.2 No ablation studies (e.g., what if we remove LEM pairing? or vary $C_n \times D_m$ ?).

4.3 Missing baselines: classical ATP systems, simpler neural architectures, retrieval methods.

4.4 No analysis of compute costs or sample efficiency.

4.5 No learning curves or training dynamics.

5. Reproducibility and availability concerns

5.1 No mention of dataset release, code availability, or benchmarking infrastructure.

5.2 Missing details: exact Lean version, prover configurations, timeout settings, hardware specs.

5.3 No appendix with example problems beyond the single counterexample.

5.4 Cannot assess problem difficulty or diversity without access.



Minor Issues

6. Methodological gaps: The "seed-data independence" claim is misleading as you still need a formalized domain. Triviality filtering is mentioned but not formally defined. Why only positive conjunctions in conclusions? This seems arbitrary. Moreover, no discussion of semantic diversity vs. syntactic diversity.

7. Limited scope of generation: Only universal quantifiers tested (existential "mentioned" for future work). Only 2 premises per problem (why would you not vary this?). Single domain (why not demonstrate transferability?). No nested constructors or complex quantifier structures.

8. Motivation for domain choice: Claim "niche domain reduces contamination" but there is no empirical verification.
Claim: "AI can help experts", but no expert evaluation or case studies. The connection between argumentation theory and transferable reasoning patterns is asserted, not demonstrated.

9. Missing comparisons: How does performance compare to human experts? How hard are these problems for classical ATPs (E, Vampire, Z3)? Can retrieval-augmented methods help? What about fine-tuning on argumentation theory?

**Questions:**

Main questions

1. On benchmark scale and utility:

1.1 Could you please provide justification for the 858 instance size? This seems orders of magnitude too small.
Have you tried scaling to $C_3​, C_4$​, or larger $D_m$​? What would prevent this?

1.2 Could you show that model performance on ArgBench correlates with performance on other reasoning benchmarks?

2. On counterexample construction:

2.1 Could you categorize the types of negative instances (e.g., by semantic property violated)?

2.2 Have you tried giving models the counterexample and asking them to verify it?

2.3 Can classical model finders (Nitpick, Nunchaku) solve these negative instances? If so, can we distill their approaches?

2.4 Are there patterns to which negative instances are "almost proven" vs. completely failed?

3. On domain generalization:

3.1 What evidence do you have that performance on abstract argumentation transfers to other reasoning domains?

3.2 How do models pretrained on math perform compared to models pretrained on logic?

4. On evaluation rigor:

4.1 Why only 1 sample per model? What is the variance with different random seeds or sampling strategies?

4.2 What are the exact hyperparameters, timeout settings, and computational budgets for each prover? Could you provide pass@k curves for k > 1?

Clarification questions

5. Methodology: How exactly is "Trivial" defined? Can you formalize this filter? Why does $C_2 \times D_1$ make sense? What happens with imbalanced ratios? What percentage of generated instances survive triviality filtering?

6. Data quality: Could you provide difficulty distribution statistics? Are there any instances that ALL provers solve? Or NONE solve? How diverse are the problems semantically (not just syntactically)?

7. Practical concerns: When will the dataset, code, and evaluation scripts be released? What are the compute requirements for regenerating the benchmark? Can this be integrated with existing evaluation frameworks (LEAN-Gym, LeanDojo)?

8. Future directions: You mention "RL training for counterexamples", have you tried this? What are preliminary results? Can you use Nitpick to automatically generate training data with counterexamples?

---

### Official Review · Reviewer_4bve · 2025-11-06

**Soundness:** 2
**Presentation:** 2
**Contribution:** 2
**Rating:** 2
**Confidence:** 4

**Summary:**

This paper proposes a benchmark for the formalization of argumentation theory —— ARGBench, which is a completely new domain. And this can reduce the risk of pre-training data leakage or contamination. Secondly, the paper introduces a Type-theoretic generation algorithm, which can construct large-scale datasets with minimal human effort. The paper evaluates four state-of-the-art provers. The results show that even the best-performing models (DeepSeek-Prover-V2 and Goedel-Prover-V2) get very low pass rates (8.16% and 5.71%, respectively). Moreover, all provers have a 0% success rate when proving negation.

**Strengths:**

1. The paper focuses on a completely new area — argumentation theory — and proposes a benchmark of sufficient scale.
2. Propose a new Type-theoretic generation algorithm.

**Weaknesses:**

1. The analysis of examples in the paper is very limited and unclear.
2. There is not much analysis of the correctness of the algorithm, nor demonstrations of it through some examples.
3. The paper claims that the algorithm can be decoupled from any specific domains, but there is not enough evidence to support this claim.

**Questions:**

1. What is this “structured type AF”? I haven’t seen such a type in mathlib.
2. The example you provided in Listing 1 is not a compilable example. What exactly are you trying to express with it?
3. The paper claims the generation algorithm is decoupled from any specific domain, could you show some cases in some specific domain (like Math)?

---

### Meta-Review · Area_Chair_RBW6 · 2025-12-13

**Summary:**

This paper presents a benchmark for automated theorem proving. The reviewers believe the paper is overall not ready yet for publication. Without a response from the authors (at all), I believe the paper needs to be further improved before publishing.

**Reviewer Scores:**

I can't predict that.

---

### Decision · Program_Chairs · 2026-01-26

Reject